# Contaminants of Emerging Concern (CECs) and Male Reproductive Health: Challenging the Future with a Double-Edged Sword

**DOI:** 10.3390/toxics11040330

**Published:** 2023-03-30

**Authors:** Daniel Marcu, Shannen Keyser, Leslie Petrik, Samuel Fuhrimann, Liana Maree

**Affiliations:** 1School of Biological Sciences, University of East Anglia, Norwich NR4 7TJ, UK; 2Comparative Spermatology Laboratory, Department of Medical Bioscience, University of the Western Cape, Private Bag X17, Bellville 7535, South Africa; skeyser@uwc.ac.za; 3Environmental and Nano Sciences Group, Department of Chemistry, University of the Western Cape, Private Bag X17, Bellville 7535, South Africa; 4Department of Epidemiology and Public Health, Swiss Tropical and Public Health Institute (Swiss TPH), 4123 Allschwil, Switzerland

**Keywords:** contaminants of emerging concern, environmental pollution, male fertility, pesticides, pharmaceuticals, semen quality, sexual development, spermatozoa

## Abstract

Approximately 9% of couples are infertile, with half of these cases relating to male factors. While many cases of male infertility are associated with genetic and lifestyle factors, approximately 30% of cases are still idiopathic. Contaminants of emerging concern (CECs) denote substances identified in the environment for the first time or detected at low concentrations during water quality analysis. Since CEC production and use have increased in recent decades, CECs are now ubiquitous in surface and groundwater. CECs are increasingly observed in human tissues, and parallel reports indicate that semen quality is continuously declining, supporting the notion that CECs may play a role in infertility. This narrative review focuses on several CECs (including pesticides and pharmaceuticals) detected in the nearshore marine environment of False Bay, Cape Town, South Africa, and deliberates their potential effects on male fertility and the offspring of exposed parents, as well as the use of spermatozoa in toxicological studies. Collective findings report that chronic in vivo exposure to pesticides, including atrazine, simazine, and chlorpyrifos, is likely to be detrimental to the reproduction of many organisms, as well as to sperm performance in vitro. Similarly, exposure to pharmaceuticals such as diclofenac and naproxen impairs sperm motility both in vivo and in vitro. These contaminants are also likely to play a key role in health and disease in offspring sired by parents exposed to CECs. On the other side of the double-edged sword, we propose that due to its sensitivity to environmental conditions, spermatozoa could be used as a bioindicator in eco- and repro-toxicology studies.

## 1. Introduction

Reproduction is a key biological event that ensures the continuation of any species [1]. Thus, species preservation and sustainable development strategies depend critically upon population dynamics and sexual reproductive health, both of which can be affected by various agents [2]. Elements interfering with reproductive processes can have profound effects on species’ evolution and the equilibrium of entire ecosystems. In this regard, the Environmental Protection Agency (EPA) has classified contaminants of emerging concern (CECs) as chemicals or materials which have a perceived, potential, or real threat to human health and the environment [3]. Such contaminants include for example pharmaceuticals, personal care products, pesticides, flame retardants, plasticizers, endocrine disruptors, surfactants, and polycyclic aromatic hydrocarbons [4,5]. Over the last two decades, various diseases and mass mortalities in marine invertebrates, mammals, and birds have been attributed to chemical exposure [6,7,8]. Human health is equally compromised as was shown by repeated studies linking long-term environmental chemical exposure to various diseases including cancer, asthma, and hypersensitivity [9,10,11]. Furthermore, evidence is mounting that such compounds are likely to interfere with human reproduction and offspring health [12].

During the past six decades, infertility rates in developing countries have increased from about 8% to 35% [13]. According to the World Health Organization (WHO) statistics, approximately 9% of couples worldwide experience fertility problems, and male fertility issues account for about 50% of these cases [14]. Male factor infertility is associated with many genetic and lifestyle factors; however, approximately 30% of cases are still considered idiopathic [14]. Various intrinsic and extrinsic factors can lead to diminishing semen quality. Examples of intrinsic physiological factors include conditions such as varicocele, metabolic disturbance, cryptorchidism, hypogonadism, hormone imbalances, and genetic aspects [15]. On the other hand, extrinsic environmental factors may include uncontrolled, prescribed, and inappropriate usage of medications, exposure to pollutants (e.g., chemicals in air, food, and water), and addictive disorders (e.g., alcoholism, smoking, and illicit drugs) [16].

Since several CECs can bind to gonadal steroid receptors, mimic steroid hormone action, and affect steroid hormone production and turnover, the decline in semen quality could likely be due to environmental (extrinsic) rather than physiological (intrinsic) factors [12,17]. Considering that male reproduction involves complex biological processes, male factor infertility is increasingly recognized as a biomarker of a male’s overall health and is associated with future disease risks including cancer, metabolic disease, and mortality [18].

Globally, poor water quality is largely determined by the level of aquatic pollution and is of great concern to public health and well-being [19]. Numerous studies have reported endocrine disruptors, pesticides, pharmaceuticals, illicit drugs, and personal care products in surface or drinking water sources [20,21,22,23,24,25]. While relative assessments indicate that traces of these compounds are present in drinking water, the majority are detected at low concentrations (ng/L to µg/L). Nonetheless, such minimum traces found in water or sediments are likely to bioaccumulate in species exposed to constant trace amounts of many different compounds. Consumption of animals exposed directly to such contaminants is becoming increasingly problematic [19]. Ojemaye and Petrik [26,27,28] reported that the levels of contaminants found in seawater, sediment, and several marine organisms (seaweed, invertebrates, and fish) may pose a threat to various trophic levels due to their high bioaccumulation factors and calculated risk quotients.

There are three objectives of this narrative review: (i) to present an overview of the existing literature regarding the effects of the selected CECs on male fertility, (ii) to assess the impact of CECs on the offspring of exposed parents, and (iii) to explore the idea of using spermatozoa as a bioindicator of environmental change as well as its potential use for future toxicology studies. Six CECs discussed in this narrative review (pesticides (atrazine, simazine, and chlorpyrifos) and pharmaceuticals (diclofenac, naproxen, and sulfamethoxazole)) were selected based on their high prevalence in the near-shore marine environment of False Bay around Cape Town [28,29] as well as in rivers [30,31], ambient air [32,33], household dust [34], and human samples in agricultural areas in the Western Cape, South Africa [35]. The effects of the selected CEC groups on male reproductive health were evaluated through a comprehensive survey of in vivo and in vitro studies across taxa, with emphasis on the two groups of contaminants. Overall, this review underscores the importance of understanding the potential impact of CECs on male reproductive health.

## 2. Origin and Distribution of CECs

While CECs have been developed for specific industrial applications and are useful for a wide range of purposes, they can also cause undesirable effects on human and animal health [4]. CECs have been released into the environment since the industrial revolution, but the quantities and varieties of CECs detected have accelerated in the last 50 years [5]. Indeed, such compounds have been consistently identified in wastewater, surface water, groundwater, and treated drinking water at low concentrations (i.e., ng/L to µg/L) [36,37].

During the past few decades, much has been revealed about the sources, transport, and biological effects of CECs in aquatic ecosystems [38]. Such contaminants are introduced into the ecosystem via two sources, namely point and non-point sources [39]. Point sources include small to large wastewater treatment plants treating sewage from municipal and industrial sources, as well as hospitals, whereas non-point sources refer to landfill leachates, surface runoff, atmospheric deposition, and agricultural applications of biosolids and manure (Figure 1) [38].

Although many treatment technologies, including activated carbon and reverse osmosis membranes, have been used for the removal of CECs, these approaches have failed to remove them [39]. In addition, degradation intermediates could be more toxic than their original compounds, which poses a great challenge to overcome [39]. Unlike conventional pollutants, CECs are rarely globally regulated [37] and can therefore present a significant risk to various organisms, ultimately affecting human health through the food web. Currently, information about co-occurrence, synergistic effects of complex mixtures, and biomagnification of CECs through different trophic levels of the aquatic food web (Figure 1) as well as its effect on individual species from different trophic levels are lacking [40,41].

Bioaccumulation of a contaminant refers to the buildup of the CECs in an organism, due to exposure through both its abiotic environment and its food sources [42]. Bioaccumulation of CECs has been observed in aquatic organisms worldwide and may be elevated in fish, for instance, due to limited intrinsic clearance mechanisms [43]. Moreover, some CEC mixtures result in greater bioaccumulation and stronger effects than that of a single CEC [38]. For example, heavy metals can accumulate in fish disrupting steroidogenesis, impairing hormone production in both sexes, and causing a reduction in the quality and quantity of gamete production [42]. Endosulfan, a polychlorinated compound used as a pest control, has also been reported to bioaccumulate in marine organisms [43], causing decreased adenylate energy charge, oxygen consumption, hemolymph amino acids, succinate dehydrogenase, heartbeat (mussel), and altered osmoregulation [44]. Benzotriazole, a corrosion inhibitor, has been shown to bioaccumulate in fish tissue and may act as an endocrine disruptor [45].

An increased awareness exists that oceans and seas can contribute to the production of feed, raw materials, and biomaterials, and seafood is widely recognized as a nutritious and high-quality food source [46]. However, seafood, similar to other types of foods, can contain harmful CECs [46]. As such, there is an increasing need for information about the presence and potential effects of any pollutants that accumulate in marine biota and the marine food web, resulting in potential contamination of seafood [46].

Even though these contaminants are usually found at low concentrations in aquatic environments, they may produce adverse short- and long-term effects over time [19]. The effects of complex mixtures of CECs on organisms are often subtle, sublethal, and indirect [38]. Multigenerational exposure to CECs in the aquatic environment may delay the occurrence of adverse effects or may result in evolutionary adaptation to historically exposed populations, making it more challenging to detect sublethal effects [38]. Currently, there is limited information about how CECs may damage organisms, but some studies have reported adverse effects involving chronic toxicity [47], endocrine disruption [48], and the development of bacterial pathogen resistance [49].

## 3. Impact of CECs on Male Fertility

Male infertility can be classified according to etiology and severity, ranging from minor changes in semen characteristics to complete spermatogenic dysfunction of the gonads. It therefore remains challenging to assess the causes of male infertility without accounting for both intrinsic and extrinsic factors which could contribute to clinical phenotypes [50]. Evaluation of semen quality is the main component for determination of male reproductive health [51]. Pharmaceuticals, personal care products (PPCPs), and pesticides have been shown to have unfavorable effects on semen quality, by negatively impacting the hypothalamic–pituitary–gonadal (HPG) axis, Sertoli and Leydig cells, spermatogenesis, steroid hormone production, and ultimately sperm function [52,53]. CECs can interfere with spermatogenesis by (i) disrupting endogenous hormone production, kinetics, and signaling pathways in the HPG axis and (ii) by disrupting the blood–testis barrier (BTB), allowing the passage of such compounds into the seminiferous tubules. Both these interferences may thus compromise the development of spermatozoa and ultimately a male’s fertility (Figure 2). It is therefore essential to evaluate the effects of CECs on male reproduction.

Table 1 [54,55,56,57,58,59,60,61,62,63,64,65,66,67,68,69] includes a summary of the effects of several CECs grouped as pesticides and pharmaceuticals on male fertility (e.g., hormone levels, testis size and structure, and sperm characteristics) in different species. Example of CECs listed in Table 1 [54,55,56,57,58,59,60,61,62,63,64,65,66,67,68,69] included both *in vivo* and *in vitro* treatments and illustrates the broad negative reproductive effects that CECs have. Overall, collective findings indicate defects in reproductive function across a wide range of marine and terrestrial animals including fish [70], birds [71], alligators, turtles, salamanders [72], mice [73], and panthers [74], when exposed to CECs. There is considerable evidence that chemical pollution commonly interferes with hormone function, leading to endocrine disruption [75]. Due to the vulnerability of hormone-receptor systems, certain endocrine disruptors affect normal reproductive functions as well as embryo development [75]. Therefore, CECs causing hormone disruption or direct damage to spermatogenesis may be responsible for changes in male reproduction (Figure 2) [13,75].

Chronic, low-dose exposures to multiple chemicals are challenging to identify, yet these are extremely prevalent [16]. Studies have shown that these exposures can have dramatic effects on both individual and population health, and interest in the cumulative and synergistic effects of such exposures on spermatogenesis and sperm function is on the rise [16]. A study assessing the reproductive health of 26,400 male workers on banana and pineapple plantations in 12 developing countries found that 24% of workers exposed to the 1,2-dibromo-3-chloropropane pesticide suffered from azoospermia, and 40.3% had oligospermia [76]. Not only did these men have compromised fertility, but only about 2.5% had fathered offspring. Because sperm production is so variable and intricate, chronic exposure may affect spermatogenesis at many levels [16]. Moreover, acute exposures to highly toxic substances can cause dramatic short-term and long-term changes in sperm characteristics [16].

In this particular review, we focused on a selected group of CECs found in False Bay, Cape Town, South Africa, including pesticides such as atrazine, simazine, and chlorpyrifos, as well as pharmaceuticals such as sulfamethoxazole, diclofenac, and naproxen. We discuss their use and potential mechanisms of action impairing a male’s reproductive success.

### 3.1. Pesticides

Pesticides comprise semi-volatile persistent organic pollutants (POPs), categorized by their use as biocides, fungicides, bactericides, insecticides, and herbicides [26,27,77]. Generally, a pesticide is defined as “any substance or combination of substances that is used to prevent or eradicate undesirable insects, including vectors of disease in animals, weeds, fungi, or other organisms in order to enhance food production, and to facilitate the processing, storage, transportation, or marketing of food and agricultural commodities” [78]. Due to the ever-increasing global population and the accompanied need for a greater food supply, these agents are expected to be more widely utilized [79]. It is estimated that in the last three decades, pesticide use has increased at least two to three times worldwide [80], highlighting that further research on their effects on public health is urgently needed.

Pesticides commonly enter aquatic environments via surface runoff and wastewater effluents from agriculture production and household use (Figure 1) and may accumulate in sediments and marine organisms [40,41,81]. Irrespective of whether they are applied intentionally or by runoff, pesticides remain in various water sources for a long time due to their chemical properties such as adsorption and solubility [82]. In general, most herbicide monitoring studies are focused on surface freshwater sources such as lakes, rivers, and reservoirs, with a particular focus on organochlorine and organophosphorus compounds [26,27,77,79].

After assessing organochlorine pesticide levels in nine fish species from Taihu Lake, China, reports suggested that consumption of more than 250 mg/d of semi-essential fatty acids from the fish could cause cancer [83]. The authors found that pesticide concentrations varied amongst specific muscle groups of the fish. Another study conducted in South Africa examined the concentration of herbicides in several organs of four wild fish species sold at Kalk Bay harbor in Cape Town [77]. The authors reported the presence of herbicides, namely simazine and atrazine, in the liver, intestines, gills, and filet. Thus, these chemicals not only pose a threat to the animals’ health and the ecosystem as a whole but also to human health as these fish species are typically consumed.

Another source of concern is the direct exposure to pesticides (air or residues) of people working in agricultural fields, as well as their children due to living on the farms. The latter group is of particular concern as they are physically not yet fully developed and have regular contact with contaminants due to frequent hand-to-mouth activities, eating more food per body weight and height, and playing in outdoor areas where potential contaminants’ residues are present. In addition, a link has also been found between pesticide exposure and a reduction in semen parameters, thus affecting fertility [78]. A study conducted on urine samples from Australian community children (aged 0 to 5 years old) found the presence of 3,5,6-trichloro-2-pyridinol (TCPY), a specific metabolite of chlorpyrifos, in each sample [84]. The study further suggested that although the “worse-case scenario” daily intake of chlorpyrifos was found to be two-fold lower than the Australian Acceptable Daily intake guidelines, the levels of metabolite detected in Australian preschoolers are higher than in other countries. In addition, recent observations of 1001 children and adolescents, as part of the cohort CapSA (described in Chetty-Mhlanga et al. [85]) of agricultural areas in the Western Cape of South Africa, indicate an overall negative trend ascribed to long-term pesticide exposure with headaches and neurocognition function reported [85,86]. Another comprehensive study, assessing 181,842 individuals performing agricultural related activities in France, reported a two-to-three-fold increased risk in central nervous system tumors in the studied population [87]. While these studies did not evaluate the reproductive organs and function of the participants, it is likely that a longitudinal study would reveal alterations in fertility, as nervous and reproductive tissues share numerous molecular mechanisms [88].

Pesticides used on fruits, vegetables, and crops can also leave potential harmful residues. As infants, children, and adults consume these foods daily, this is a major concern. Residue levels were found to be above the WHO’s lower limit in vegetables commonly sold at six different markets in Lagos, Nigeria [89]. It was also noted that exposure to vegetable pesticides could occur either at storage sites or in the field. Not only are such pesticides sprayed upon the vegetables, but their continuous use is likely to result in leakage into the soil and subsequent uptake by the plants. According to the authors, such vegetables could potentially cause bioaccumulation and health risks. Furthermore, after examining pesticide residues in vegetables and fruits from Qatar, Al-Shamary et al. [90] found insecticide concentrations of imported fruit and vegetables to be above the maximum acceptable residue levels. Due to the poisonous nature of pesticides, regulatory bodies must be vigilant in their oversight of pesticides and rely on science to develop appropriate protocols to maintain an equilibrium between beneficial use and adverse consequences [91].

Despite the growing evidence of the various routes of pesticide exposure to humans, the effects of these CECs on male reproductive health and sperm functionality remain elusive. Many agricultural and non-agricultural pesticides are hormonally active, including organophosphates, pyrethroids, triazines, azoles, and carbamates. As such, they have the potential to interfere with the endocrine system which controls various important reproductive processes. Previous reviews have demonstrated significant associations between pesticide exposure and diminished sperm quality in humans [78,92]. Research suggests that occupational exposure to pesticides can result in male reproductive system pathology, such as damage to testes, impaired spermatogenesis, and reduced semen quality [55].

#### 3.1.1. Atrazine

One of the most used chlorotriazine herbicides is atrazine, which persists in water and soil for extensive periods due to its long half-life (>60 days) [55,93]. Despite being banned in the European Union and restricted in other countries, atrazine is still found in water at levels exceeding recommended limits (US Environmental Protection Agency (EPA) = 3 μg/L, European Union = 0.1 μg/L) [57]. In accordance with US EPA tolerances and drinking water, recent reports indicate that acute dietary exposures to atrazine in humans range from 0.234 to 0.857 μg/kg/day, and chronic dietary exposures range from 0.046 to 0.286 μg/kg/day [57]. Various investigations have indicated atrazine as a potent endocrine disruptor, which may affect reproduction in mammals, birds, amphibians, reptiles, and fish [57]. Atrazine passes biological barriers, such as the blood–brain barrier (BBB), targeting the HPG axis and the BTB, causing oxidative stress, inflammation, mitochondrial dysfunction, and apoptosis in the exposed cells [93]. Low doses of in vitro exposure to atrazine (0.1 or 1 μM) or its major metabolite diaminochlorotriazine (DACT; 1 or 10 μM) have been demonstrated to disrupt sperm membranes and acrosome integrity and functionality, as well as mitochondrial function in bovine spermatozoa [55].

#### 3.1.2. Simazine

Simazine have been found both in surface and ground water sources as well as in food products, which can lead to human exposure through consumption [94]. An additional risk of occupational simazine exposure has been reported through skin contact during mixing or application of this pesticide [94]. Across Europe, simazine was one of the triazine herbicides most frequently detected above regulatory levels [82]. Triazines such as atrazine and simazine display similar modes of action and have been implicated in a variety of cancers according to different studies [95]. In addition, long-term consumption of high doses caused tremors, damage to the testes, kidneys, liver, and thyroid, and decreased sperm production in laboratory animals [95]. Due to its endocrine disruptor-like characteristics and large production volume, simazine was included in a final list of chemicals tested in the US EPA’s endocrine disruptor screening program in 2009 [96]. Researchers reported that male offspring exposed to simazine proved to have decreased body weight, testicular size, and epididymis mass, increased testicular apoptosis, and low sperm counts [96,97]. Simazine is thought to act by downregulating genes such as those involved in the relaxin pathway, including nitric oxide synthase 2 (Nos2) and Nos3 [96,97]. Thus, simazine results in the reduction of nitric oxide (NO) production in rat Leydig cells in vitro [96,97] and alters the expression of genes that are critical for regulating apoptosis and steroidogenesis [98].

#### 3.1.3. Chlorpyrifos

Another widely used pesticide in both agriculture and industrial sectors across the world is chlorpyrifos, globally adopted due to its broad-spectrum effectiveness against insects [99]. After reviewing scientific studies, the EPA concluded that chemical residues on or in food are unsafe based on the cumulative exposure to chlorpyrifos [91,100]. Despite this evidence, no revocations or cancellations of chlorpyrifos registrations were made [91,101]. Since then, several reports have indicated that, in addition to its acute toxicity, hepatotoxicity, nephrotoxicity, neurotoxicity, and developmental toxicity, chlorpyrifos affects male reproduction [58]. However, detailed investigations regarding chlorpyrifos reproductive toxicity and mechanisms are lacking [58].

Observations have suggested that exposure to chlorpyrifos may increase intracellular reactive oxygen species, thereby leading to oxidative stress and damage in cells [58]. An in vivo study on mice fed 3.0 mg chlorpyrifos/kg body weight for 20 weeks demonstrated significantly decreased sperm counts, serum testosterone, and gonadotropin levels and enzyme activity related to spermatogenesis [102]. Zhang et al. [58] exposed mouse spermatozoa to 25 μg/mL chlorpyrifos and after 1 h observed significantly decreased motility and mitochondrial membrane potentials with an increase in reactive oxygen species (ROS). Interestingly no significant effect was observed on sperm viability [58].

### 3.2. Pharmaceuticals

Pharmaceuticals and personal care products (PPCP) have a wide range of applications including being used as antibiotics, hormones, antimicrobial agents, and synthetic musks [103]. In human and veterinary medicine, most pharmaceuticals are used to prevent or treat infectious or lifestyle diseases, whereas large amounts are also used in agriculture to promote fruit growth and in livestock and fish farming to promote growth and prevent disease [69]. Since pharmaceuticals are biologically active substances that can interfere with the biochemical and physiological processes of non-target organisms when they ends up in water resources (Figure 1), they are recognized as being CECs [104]. A large number of PPCPs are excreted as the parent compound or as its metabolites, which flow into wastewater treatment plants [48]. While the concentration of some of these compounds can be controlled or reduced by facility-specific treatment practices, many CECs are not properly removed or are discharged into surface waters, including streams, estuaries, or open marine waters due to secondary bypass or combined sewer overflows [48,104]. Additionally, considering that approximately 70% of pharmaceuticals consumed by humans are ionized weak bases, more research is needed to understand pH influences on the bioavailability and toxicity of ionized pharmaceuticals [43].

In aquatic organisms, pharmaceuticals are accumulated as a result of two primary processes, namely direct partitioning from the abiotic environment (bioconcentration) and trophic transfer (dietary exposure) [105]. The majority of pharmaceuticals are more polar and less hydrophobic than most CECs and thus do not preferentially associate with sediment or tissue [48]. Even so, they can be bioaccumulated through ventilation, ingested water, and prey and therefore may interact with receptor targets, causing pharmacological effects in non-targeted organisms when concentrations are high enough [48].

Several types of PPCPs have been found in water, sediments, and fish in the Mediterranean River Basins [40,41] and in South Africa [28,77,106]. Those detected in fish samples included anti-inflammatories (diclofenac), psychiatric drugs (citalopram, carbamazepine, and venlafaxine), and β-blockers (clopidogrel, carazolol, sotalol, and propranolol). Interestingly, the most frequently detected PPCP was diclofenac [28,40,41,106]. According to a recent study of 12 fish species from a variety of families, more than 65% of drug targets had orthologues in humans [107], suggesting that many of the drugs metabolized and bioaccumulated in fish might also negatively affect humans.

#### 3.2.1. Sulfamethoxazole

Pharmaceuticals of emerging concern in African surface waters include antiretrovirals and antibiotics [108]. Antibiotics rank as one of the most commonly used and consumed pharmaceutical classes, with low levels widely detected in sewage treatment plants effluent, surface water, groundwater, and drinking water [69]. Trimethoprim/sulfamethoxazole has been the choice of antibiotic therapy for the last 30 years due to its effectiveness against both gram-negative and gram-positive bacteria [109]. However, as a result of the sequential blockade of multiple steps involved in microbial folate synthesis, trimethoprim/sulfamethoxazole can inhibit the formation of purines and, ultimately, DNA [68,110]. In addition, the compound can also cross the placenta and thus harm neonates due to its folic acid antagonist properties [109]. Low folate levels in seminal plasma are reported to be associated with increased sperm DNA damage [111].

Oputiri and Elias [68] found that male rats orally treated with sulfamethoxazole/trimethoprim (22.4/4.6 mg/kg) showed decreased semen quality (sperm count and motility) associated with increased reactive oxygen species (ROS), histological testicular damage, and abnormal spermatozoa. Salarkia et al. [111] observed that adult male Wistar rats treated in vivo with trimethoprim/sulfamethoxazole (30, 60, and 120 mg/kg/day) for either 14 or 28 days presented with significantly decreased sperm counts and percentages of motility and viability. Moreover, a study conducted by Hargreaves et al. [112] indicated that at low concentrations, trimethoprim/sulfamethoxazole did not affect human sperm movement; however, at 500 µg/mL, it reduced the movement by 34% [113]. Thus, exposure to trimethoprim/sulfamethoxazole can potentially decrease male fertility through possible inhibition of meiosis of primary spermatocytes, direct destruction of spermatozoa, interfering with energy production and mitochondria, or decreasing folate levels in seminal plasma [68].

#### 3.2.2. Non-Steroidal Anti-Inflammatory Drugs

In South Africa, non-steroidal anti-inflammatory drugs (NSAIDs) have historically been the most consumed category of drugs, followed by antibiotics [114]. NSAIDs are used to treat various illnesses either alone or in conjunction with other pharmaceuticals [114]. NSAIDs such as acetylsalicylic acid, ibuprofen, and naproxen are commonly available over the counter, thereby increasing their prevalence in the environment. Ibuprofen, naproxen, diclofenac, ketoprofen, and fenoprofen are reported to be the most prominent NSAIDs found in aquatic environments of South Africa [114] and should be considered as potential CECs.

NSAIDs inhibit the non-selective activity of cyclooxygenase (COX)-1 and -2 isoforms, decreasing the catalysis of prostaglandin (PG) biosynthesis from phospholipid arachidonic acid [115]. These compounds are widely used for their analgesic, antipyretic, and anti-inflammatory properties [116]. The PG family consists of lipid-signaling molecules derived from polyunsaturated fatty acids and are involved in a variety of biological processes, including fertilization [117]. As a result, they regulate human reproduction, neurological function, cancer progression, and inflammation and serve as short-lived, local hormones [117]. An association has been suggested between PGs and sperm motility as PGF1𝝰 binds with high affinity to the sperm calcium channel (CatSper), which in mammalian spermatozoa is crucial for generating hyperactivated motility and therefore fertilization [117]. In addition to their role in the regulation of testicular functions, PGs have been suggested to exert stimulatory as well as inhibitory effects on spermatogenesis [66].

Various water sources, such as drinking and groundwater, have been found to contain naproxen, ranging in concentrations from ng/L to μg/L [118]. In spite of their low concentration, NSAIDs have the potential to cause adverse effects on non-target organisms in long-term exposure or while mixed with other drugs [118]. As a result, naproxen and its byproducts can be harmful to living organisms, including humans [119]. Uzun et al. [66] treated male rats with naproxen and meloxicam and observed a significant decrease in sperm count and motility, with induced damage of seminiferous tubules, without any effect on plasma hormone levels. According to the study, reproductive toxicity may be caused by the inhibition of PG synthesis, whereas oxidative stress may also play a key role [66]. The latter effect was confirmed in a study by Ahmad et al. [120] that suggested naproxen to be a potential genotoxic agent. Oral naproxen administration to male Wistar rats for 14 days resulted in biochemical imbalances and induced oxidative stress, which weakened the integrity of the cells [120].

Diclofenac, one of the most prescribed NSAIDs, is widely used to treat pain and inflammation, but it has been closely associated with adverse effects on avian fauna and raised environmental concerns [121,122]. In relation to other commonly used NSAIDs, diclofenac is reported to be about three to 1000 times more effective on a molar basis and in its ability to inhibit the activity of COX [104]. Despite acting as an anti-inflammatory, diclofenac has also been associated with decreasing antioxidant indices and thus may induce oxidative stress in cells [123]. Vyas et al. [121] administered diclofenac (0.25 mg/kg, 0.50 mg/kg, and 1.0 mg/kg) to male rats for 30 days and observed significant decreases in weight of the testis, epididymis, ventral prostate, and seminal vesicles [121]. A dose-dependent decrease was also observed in sperm count, density (in epididymis and testis), motility, and testicular cell population dynamics [121]. Mousa et al. [124] observed decreased sperm counts, individual sperm motility, and viability as well as depleted concentrations of reduced glutathione in testicular tissue, decreased testosterone levels, and alteration in testicular histological features in rats treated with diclofenac (2.5 mg/kg body weight) four times/week for 8 weeks.

Diclofenac may thus induce its negative effects on both qualitative and quantitative measures of spermatozoa [122]. These consequent effects may be attributed to reduced levels of gonadal hormones, decreased antioxidant defense mechanisms, increased oxidative stress, altered concentrations of nitric oxide that are required to maintain normal sperm physiology, and reduced synthesis of PGs [122]. However, further investigation on these possible mechanisms and effects are required, especially for in-depth sperm functional characteristics related to fertilization success.

## 4. Impact of CECs on Offspring and Potential Mechanisms

Since their introduction in the environment more than 50 years ago, CECs have been found to not only affect the health of exposed individuals, but also of subsequent generations. For instance, environmental exposure to endocrine disruptors has been linked to headaches [86] and neuro-developmental disorders in upcoming generations [125,126,127]. Spermatozoa have been reported to deliver non-genetic factors into the oocyte [128]. The mechanisms underlying such transfer include the binding of methyl/acetyl groups to DNA, modification of sperm histones affecting gene expression [129], and transfer of RNA families or proteins present in spermatozoa into the zygote [130]. These heritable changes, known as epigenetic marks, do not necessarily involve the alteration of the DNA sequence but rather promote alterations in gene function without changing the underlying genome. Such modifications result in altered gene expression in response to environmental factors without affecting the DNA code itself. Given their role in controlling gene expression, it is likely that paternal conditions, such as exposure to CECs, would affect epigenetic patterns in the offspring. Whilst evidence in humans is often technically challenging to obtain, in animal models, on the other hand, evidence is mounting and emphasizes that the perturbation of epigenetic marks might account for infertility cases and diseases in the offspring.

Early embryonic development is critically sensitive to epigenetic manipulations by exposure to environmental contaminants. In exposed females, CECs can be transferred into the breast milk and cross the blood–placental barrier, compromising the development of the progeny. In fact, rodent studies suggest that exposure of mothers to therapeutic doses of paracetamol not only negatively affects the testosterone levels of male pups [131] but also compromises the ovarian reserve of female offspring [132]. Exposure to diclofenac, particularly, in pregnancy seems to dramatically affect the total number of Sertoli and Leydig cells in the rat male offspring [133]. In utero exposure to other forms of CECs including the pesticide permethrin, insect repellent *N*,*N*-Diethyl-meta-toluamide, plastic additives such as bisphenol A and phthalates, dioxin, and jet fuel was also shown to promote male germline alterations in F3 progeny [134]. In addition, exposure to the pesticide atrazine also induces epigenetic transgenerational inheritance. Two generations of mice following parental exposure presented sperm epigenetic alterations, with only F2 and F3 having a higher incidence of early onset puberty and testis disease, respectively [135]. Such evidence suggests that parental exposure to CECs affects not only the parents themselves but also leaves profound signatures in the germline, compromising the health of the subsequent generations.

Most recent preliminary in vitro studies in humans show that exposure, at human therapeutically relevant levels, to acetaminophen and ibuprofen causes loss of germ cells and modification in their epigenetic patterns [136]. Furthermore, human testis cultured with therapeutic doses of paracetamol presented up to a 30% reduction of testosterone after 24 h of exposure [137]. Further experiments using human fetal testis in a xenograft model reveal a 45% reduction in plasma testosterone and 18% reduction in the seminal vesicle weight after 7 days of therapeutic exposure to paracetamol [138]. Whilst such work highlights the physiological impact of CEC exposure on the male reproductive system, recent molecular work confirmed the molecular signatures left by CEC exposure. A recent study performed epigenetic analysis from cord blood samples from children exposed for more than 20 days to paracetamol in utero and found significant differences in DNA methylation levels compared to the control group [139]. The study further suggested that several methylated genes were linked to attention-deficit/hyperactivity disorder (ADHD), oxidative stress, neurodevelopment, and neurotransmission. In line with this evidence, another study identified several genome-wide differentially methylated regions (DMRs) in sperm produced by men under low and high exposure to the insecticide 1,1-dichloro-2,2-bis(p-chlorophenyl)ethylene (DDE). Elevated exposure to DDE was suggested to be associated with DMRs in sperm, and those regions are enriched for genes involved in neurological functions including susceptibility to autism spectrum disorders, schizophrenia, and bipolar disorder [140]. Although these studies have only focused on the direct exposure of selective groups of CECs, cumulative effects and bioaccumulation of these compounds through the different sources have the potential to contribute to subtle changes at the level of the epigenome and are thus likely to be manifested through various phenotypic forms in the germline and offspring.

## 5. Management, Detection, and Possible Treatment of CEC Exposures

Increasing urbanization has resulted in increased anthropogenic activities, economic growth, urbanization, industrialization, and exploitation of natural resources, leading to massive waste management problems, disposal problems, and the emergence of various contaminants [141]. Through various pathways, CECs contaminate soil, water, and air, affecting the environment and human health (Figure 1) [141]. A variety of analytical techniques have been employed to detect and quantify inorganic and organic contaminants in aqueous matrices; however, the presence of contaminants in water has demonstrated that current quality controls cannot detect or treat pollutants that are present [142]. Nevertheless, detection techniques still include chromatographic, spectroscopic, electrochemical, and colorimetric titration amongst others. Each of these methods weigh in with various advantages and disadvantages as critically discussed by Warren-Vega et al. [142].

Urban wastewater treatment plants (WWTPs) release organic contaminants into aquatic systems, thereby making WWTPs a major land-based pollution source of waterbodies. The removal efficiency of organic contaminants varies substantially as a result of a number of factors, including the difference in operating conditions of the treatment plants, the structural diversity of the organic contaminants, and their chemical and physicochemical properties [143]. As a result, conventional secondary (e.g., activated sludge process) and tertiary treatments (e.g., filtration and disinfection) in WWTPs are not effective in removing most CECs from the water [144]. More recently, the removal of CECs from wastewater has been investigated by using a range of new advanced treatment methods, including those consolidated (activated carbon adsorption, ozonation, and membrane filtration) and those not as intensively implemented, such as advanced oxidation [144].

As the detection and eradication of CECs proves to be problematic, the question remains whether exposure to CECs causes irreversible harm to human health, especially reproductive health. In this regard, it becomes clear that the chemical nature of the CECs, in addition to the period in an organism’s lifespan during which it is exposed, determines the duration of effects. For example, in external fertilizers, exposure to diclofenac for 14 days has led to significant reduction in fecundity and fertility; however, after being returned to clean water their reproductive ability was gradually restored [145]. In contrast, in mammals, the reproductive effects of early life exposures to CECs in utero are much more pronounced later in life, even without subsequent exposure [146]. In addition, disruptions in the HPG axis in young elephant bulls was shown to lead to a dramatic reduction in testis size, sperm production, and testosterone concentrations even years post puberty interference [147]. Consequently, it is likely that in humans, exposure to CECs during critical stages of development affects reproduction irreversibly. Although preventative strategies to limit exposure to such compounds are difficult to be implemented due to their ubiquitous presence in everyday life, a recent study found that selenium supplementation likely counteracts the reactive oxygen species and DNA damage in sperm induced by Ibuprofen exposure [148]. Nonetheless, a randomized control trial suggests that antioxidant supplementation does not improve sperm function in infertile men [149]. Therefore, the latter study speculates that if exposure to CECs dramatically affects sperm quality, compromising a man’s ability to father offspring, then antioxidant supplementation would not alleviate the harmful effects of CECs. Control measures and mitigation strategies should therefore focus on limiting human exposure to CECs through appropriately identifying and eliminating the compounds from the possible sources as well as imposing penalties for using such compounds in agriculture and manufacturing, especially when alternatives are available. In addition, establishing informational campaigns to increase public awareness about the emergence of CECs and their consequences has the promise to drive legislative action and limit the reproductive effects currently reported.

## 6. Role of Spermatozoa in Toxicology Applications

Bioindication has been one of the major directions of environmental research, and ecologists are interested not only in pollution levels but also in how organisms react to xenobiotics [150]. It is critical to use bioindicators to detect, evaluate, and monitor changes in the environment, both positive and negative [151]. A variety of factors must be considered when choosing biological indicators for bioassays, including its sensitivity and reliability, distribution, and environmental relevance, as well as availability throughout the year. Moreover, an accurate, predictable, and reliable endpoint is required for toxicity testing [152].

In toxicity screening, many validated in vivo and in vitro models are used to identify and predict the potential harmful effects of anthropogenic chemicals. However, the development and implementation of new in vitro techniques for ecotoxicology and environmental risk assessment are paramount due to the fact toxicological studies on animals are costly, require many specimens, and can raise ethical concerns [153]. Besides reducing the number of animals used in toxicity testing, in vitro model systems have several other advantages, including a reduced maintenance cost, a smaller amount of chemicals required, shortening of the time required for tests, and increasing the throughput for evaluating many chemicals and their metabolism [154].

Spermatozoa have the potential to meet many of the aforementioned requirements as an in vitro toxicological model [155]. As highly specialized cells, spermatozoa have a unique compartmentalized structure which enables them to perform a diverse range of biological functions (movement, cell recognition, secretion, and membrane fusion) required for fertilization [156]. As such, spermatozoa have measurable characteristics that respond to toxicants in a dose-response manner [155], particularly in relation to the effects of xenobiotics on biological membranes [156]. Moreover, spermatozoa are inexpensive to produce and easy to obtain and manipulate in laboratory environments. Displayed in Table 2 [157,158,159,160,161,162,163,164] are several examples of investigations (from 2006 to 2021) using spermatozoa from various phyla as toxicological in vitro models and assessing various reliable endpoints (sperm viability, kinematics and motility, and DNA fragmentation amongst others) for toxicity effects. Studies displayed in Table 2 [157,158,159,160,161,162,163,164] were selected on the basis of including in vitro sperm exposure to various CECs.

A study in *G. caespitosa* assessed the effects of four heavy metals (Cu, Zn, Pb, and Cd) on spermatozoa and found that these contaminants affect sperm density and fertilization success [157]. Another study in rainbow trout found that exposure of sperm to sublethal concentrations of two heavy metals (Hg^2+^ and Cd^2+^) alters sperm motility and larvae hatching rates [158]. A more recent study in mice evaluated the effects of sperm exposure to temephos and concluded that sperm function and metabolism, fertilization rates, and blastocyst formation rates were significantly decreased [160]. Human spermatozoa have also been shown to be responsive to environmental pollutants, serving as a powerful alert of the detrimental effect that environmental contaminants have on human health [165]. Hardneck et al. [166] were able to determine the threefold concentrations for the harmful effect of CuSO_4_ and CdCl_2_ on human sperm by calculating IC50 values, suggesting that spermatozoa could be used as a potential bioindicator for heavy metal toxicity. Vollmer et al. [167] developed human spermatozoa-based toxicity testing (HSTT) for detecting single substance-induced toxicity by using sperm motility and viability as a screening tool to identify in vivo toxins, which could possibly also be applied for environmental toxins.

The use of male gametes from plants in ecotoxicology has been established for a number of years. Studies revealed that the use of pollen to detect pesticides is beneficial for environmental monitoring [168] as it provides important insights of the consequences of environmental change in the adult plant, due to gametocytes expressing many genes that are known to correlate with sporophyte fitness [169]. Identification of pollen that is more resistant to stress correlates with more resistant crop plants [170]. The use of fish sperm for toxicity tests is another example, since fish gametes and embryos are highly sensitive to the toxic effects of water contaminants [171].

Furthermore, the human genome consists of approximately 25,000 genes, of which 10% are related to reproduction [18,172]. It is highly likely that these genes have overlapping pathways and functions with those found in a variety of cell types and organ systems. Research suggests that there is a considerable gene overlap between the testes and brain [88]. It may therefore be plausible to infer the potential outcome of CECs on other cell types by analyzing the effects on various sperm functional characteristics, and the development of spermiotoxicity assays could provide a faster alternative to traditional in vivo assays.

## 7. Conclusions

Spermatogenesis requires precise regulation of somatic cell populations which are highly sensitive to both extrinsic and intrinsic factors. Exposure to CECs such as atrazine, chlorpyrifos, simazine, diclofenac, naproxen, and sulfamethoxazole may affect male fertility and overall health through various mechanisms such as direct damage to cell structure, acting as an EDC and affecting hormones, causing DNA fragmentation and alterations, gene mutations, and epigenetic effects through altering gene expression. These extrinsic sources therefore interfere with intrinsic physiological processes that ultimately affect an organism’s health. Although CECs have a wide range of negative effects on the male reproductive system, this could be seen as an advantage for the potential use of spermatozoa as a bioindicator of toxic environments for a vast range of contaminants—a double-edged sword for addressing CECs in future studies. Having a comprehensive understanding of the effects of CECs on the functional characteristics of human spermatozoa can also aid in identifying idiopathic fertility cases and may lead to the development of more individualized treatment plans for patients undergoing ART. Furthermore, identifying the concentrations or combined mixtures at which these compounds may cause reproductive toxicity can assist in the development of legislative measures that monitor and regulate CECs.

## Figures and Tables

**Figure 1 toxics-11-00330-f001:**
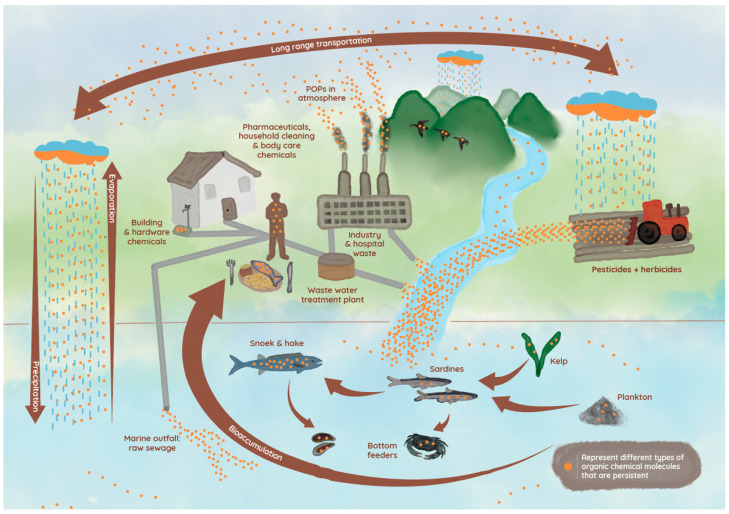
Origin and distribution of persistent CECs in the environment. Human activities and the associated chemicals developed for better quality of life and modern living conditions are the primary sources of contaminants entering natural ecosystems. Various anthropogenic sources of CECs contribute to their widespread occurrence. Examples of CEC sources include agricultural use of pesticides and herbicides, industrial and hospital waste, wastewater treatment plant effluent, and building and hardware chemicals. POPs = persistent organic pollutants. Image source www.waterstories.co.za (accessed on 1 April 2022).

**Figure 2 toxics-11-00330-f002:**
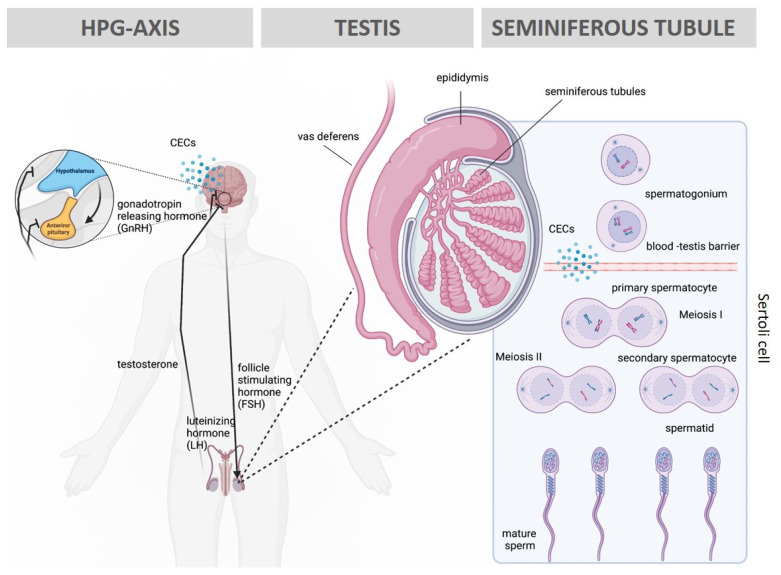
Overview of the spermatogenesis process and the levels of CECs interference. In the brain, CECs can disrupt gonadotropin releasing hormone (GnRH) production and release and interfere with the hypothalamic–pituitary–gonadal (HPG) axis. In the testis, CECs can disrupt the blood–testis barrier between the Sertoli cells, allowing the passage of contaminants into the seminiferous tubules, affecting meiotic spermatocytes and haploid spermatids. Figure generated using Biorender.

**Table 1 toxics-11-00330-t001:** The effect of various contaminants of emerging concern (CECs) on male fertility of various species.

Chemical		Concentrations	In Vivo/In Vitro	Study Population	Duration	Effects	Study
Pesticides	Atrazine	5 mg/kg bw/day	In vivo	Pregnant female mice and male pups after gestation	Early gestation day 9.5 until 12 or 26 weeks of age	↓ Epididymal sperm concentration	Harper et al. (2020) [54]
						↑ Altered steroidogenic gene expression	
						↓ Cells within the preimplantation embryo	
	Atrazine and diamino chlorotriazine	ATZ (0.1 or 1 μM) and DACT (1 or 10 μM)	In vitro	Bull semen	During cryopreservation (3 hrs) and during capacitation (4 hrs)	↓ Sperm vitality	Komsky-Elbaz et al. (2019) [55]
						↑ MMP	
						↓ Ca^++^ ionophore-induced AR	
	Atrazine	0.5, 25, and 50 mg/kg bw	In vivo	Young adult male mice	3 days	↓ Testis weight and gonadosomatic index	Abarikwu et al. (2021) [56]
						↑ Abnormal histology of gonads	
						↓ Testosterone levels and production	
						↑ Impaired spermatogenesis	
						↓ Leydig cell viability	
	Atrazine	0.1 mg/kg, 1 mg/kg, and 10 mg/kg of bw	In vivo	Adult male rodents	21 days	↓ Total and prog mot	Saalfeld et al. (2018) [57]
						↓ Sperm membrane integrity	
						↑ Membrane fluidity	
						↓ Mitochondrial functionality and acrosome integrity	
	Chlorpyrifos	Dietary CPF: 1 mg/kg/day, 3 mg/kg/day, or 12 mg/kg/day. Intraperitoneal CPF injection: 3 mg/kg/day, 6 mg/kg/day, or 12 mg/kg/day. Gavage CPF: 12 mg/kg/day and 25 mg/kg/day. In vitro: 25 μg/mL for sperm and 50 μM or 25 μM for cell lines	In vivo and in vitro	Male mice and germ cell culture	Dietary CPF: 80 days. Intraperitoneal CPF injection: 15 days. Gavage CPF: 35 or 70 days. In vitro: sperm 1 hrs, cell line 12 or 24 hrs	↓ Expression steroid hormone synthesis-related genes.	Zhang et al. (2020) [58]
						↓ Weight of gonads and associated structures	
						↑ Protein expression of Caspase3	
						↓ Sperm density and prog mot and linear movement	
						↑ ROS	
						↓ MMP	
						↓ Cell line normal morphology and viability	
	Chlorpyrifos, imidacloprid, and cypermethrin	5 mg/kg bw CYP, 9 mg/kg bw IMC, and 1.9 mg/kg bw CPF	In vivo	Male rats	5 times per week for 1 month	↑ Testis weight	Abdel-Razik et al. (2021) [59]
						↓ Epididymis and prostate gland weights	
						↓ Sperm counts, moti, viability and normality	
						↑ ROS, lipid peroxidation and testis protein carbonyl content	
						↓ Serum testosterone, LH and FSH levels	
						↓ Spermatogenesis	
						↑ Abnormal histology of gonads and associated cells	
	Chlorpyrifos	1 mg/kg	In vivo	Adult male rats	60 days	↓ Weight of gonads and associated structures	Khalaf et al. (2022) [60]
						↓ Sperm count, sperm mot and prog mot	
						↑ Dead and abnormal sperm	
						↓ Serum testosterone, FSH, and LH levels	
						↑ DNA laddering	
	Chlorpyrifos	37 mg/kg/bw	In vivo	Male rats	Sampling on days 5, 15, 30, and 45	↓ Testicular weight	Babazadeh and Najafi (2017) [61]
						↓ Sperm count, viability and mot	
						↓ Leydig cells	
						↑ Abnormal histology of gonads and associated cells	
						↑ Immature sperm and DNA damage	
Pharmaceuticals	Atorvastatin, sildenafil citrate, gemfibrozil, ibuprofen, atenolol, ofloxacin, carbamazepine, bezafibrate, and diclofenac	Atorvastatin (13 ng/mL), sildenafil citrate (26–25 ng/mL), gemfibrozil (380 ng/mL), ibuprofen (92 ng/mL), atenolol (241 ng/mL), ofloxacin (50 ng/mL), carbamazepine (310 ng/mL), bezafibrate (57 ng/mL), and diclofenac (180 ng/mL)	In vitro	Men aged 20–30 years	15, 30, and 45 min	↓ Sperm mot	Rocco et al. (2012) [62]
						↑ Genomic damage	
						↑ Apoptotic cells and DNA fragmentation	
	Indomethacin, diclofenac sodium, tolmetin, acetylsalicylic acid, resveratrol, and NS-398	0 to 15 mM	In vitro	Turkey toms	5 min	↓ Sperm mot	Kennedy et al. (2003) [63]
	Diclofenac	10 mg/kg	In vivo	Male rats	30 days	↓ Serum testosterone, LH and FSH	El-Megharbel et al. (2021) [64]
						↓ Sperm mot and count	
						↓ Testicular tissue antioxidant defence enzymes	
	Ibuprofen	25 and 50 μg/L	In vivo and in vitro	Mature male striped catfish	4 months	↓ Prog and total mot, rapid and medium speeds	Gallego-Ríos et al. (2021) [65]
						↑ Slow speeds and immotile sperm	
						↓ VCL, VSL and VAP, LIN and STR, ALH, WOB and BCF	
	Naproxen and meloxicam	Naproxen (10 mg/kg) and meloxicam (1 mg/kg)	In vivo	Male rats	35 days	↓ Sperm mot and count	Uzun et al. (2015) [66]
						↓ Prostaglandins and ROS defence enzymes in testis	
						↑ Abnormal histology of gonads and associated cells	
	Lincomycin-spectinomycin and sulfamethoxazole-trimethoprim	Lincomycin-spectinomycin injected 0.1 mL/kg bw and sulfamethoxazole-trimethoprim orally administered at 0.12 mL/kg bw	In vivo	Rams	Intramuscular injections once daily for 3 days and oral administration twice daily for 3 days	↑ Serum and semen hyaluronidase activity	Tanyildizi et al. (2003) [67]
						↓ Sperm count	
						↑ Sperm mot	
	Lopinavir-ritonavir and sulfamethoxazole-trimethoprim	22.4/4.6 mg/kg of sulfamethoxazole-trimethoprim, 22.8/5.8 mg/kg of lopinavir-ritonavir, and combined doses of sulfamethoxazole-trimethoprim + lopinavir-ritonavir	In vivo	Male rats	2–8 weeks	↑ Testicular MDA	Oputiri and Elias (2014) [68]
						↓ SOD	
						↓ Sperm mot and count	
						↑ Abnormal sperm morphology	
						↑ Abnormal testicular histology	
	Norfloxacin and sulfamethoxazole	Norfloxacin: 0.0032, 0.016, 0.08, 0.4, 2, and 10 mg/L. Mixture of norfloxacin and sulfamethoxazole: 0.0016 + 0.008, 0.008 + 0.04, 0.04 + 0.2, 0.2 + 1.0, 1.0 + 5.0, and 5.0 + 25.0 mg/L, respectively.	In vivo	One year old male goldfish	7 days	↑ DNA damage of the gonads	Liu et al. (2014) [69]

Abbreviations: ↓, decrease or negatively affected; ↑ increased or positively affected; ALH, amplitude of lateral head displacement; AR, acrosome reaction; AZT, atrazine; BFC, beat cross frequency; bw, body weight; COX, cyclooxygenase inhibitors; CPF, chlorpyrifos; CYP, cypermethrin; DACT, diamino chlorotriazine; DMSO, dimethyl sulfoxide; DNA, deoxyribonucleic acid; FSH, follicle stimulating hormone; GEH, germinal epithelium height; GPx, glutathione peroxidase; GSH, glutathione; hrs, hours; IMC, imidacloprid; LH, luteinizing hormone; LIN, linearity; min, minutes; MDA, malondialdehyde; MMP, mitochondrial membrane potential; Mot, motility; PGE, prostaglandin; Prog, progressive motility; RI, repopulation index; ROS, reactive oxygen species; SOD, superoxide dismutase; STD, seminiferous tubules diameter; STR, straightness; TDI, tubular differentiation index; VAP, average path velocity; VCL, curvilinear velocity; VSL, straight-line velocity; WOB, wobble.

**Table 2 toxics-11-00330-t002:** Studies using spermatozoa as a bio-indicator to evaluate the effect of various environmental contaminants on male fertility of various species, including examples of exogenous and endogenous fertilization strategies.

Species	Endogenous/Exogenous	Environmental Contaminant	Concentrations	Duration	Effects	Study
Sydney worm *(Galeolaria caespitosa)*	Exogenous	Heavy metals (Cu, Zn, Pb, and Cd)	Cu (12–33) Zn (160–550), Pb (560–1500), and Cd (4900–6100) μg/L	30 min	↓ Fertilization rate	Lockyer et al., 2019 [157]
Rainbow trout (*Oncorhynchus mykiss*)	Exogenous	Hg^2+^ and Cd^2^	1, 10, 100 mg Hg^2+^/L and 10, 100, 500 mg Cd^2^	4 and 24 h	↓ Viability	Dietrich et al., 2010 [158]
					↓ Kinematics	
					↑ DNA fragmentation	
Yellow-tailed lambari *(Astyanax altiparanae)*	Exogenous	Aluminium	0.05, 0.1, 0.3, and 0.5 mg/L	50 sec, 10 and 30 min	↓ Membrane vitality	de Assis et al., 2021 [159]
					↓ Mitochondrial activity	
					↓ Mot and kinematics	
Mouse *(Mus musculus)*	Endogenous	Temephos	0.1, 1, 10, and 100 mM	90 min	↓ Sperm mot, medium and rapid sperm mot, prog mot, kinematics	Kim et al., 2020 [160]
					↑ Live AR	
					↓ Live Capacitated	
					↓ ATP levels	
					↓ PKA activity and tyrosine phosphorylation	
					↓ Fertilization rate	
Buffalo (*Bubalus bubalis*)	Endogenous	Cadmium, lead, chlorpyrifos, and endosulfan	0.005, 0.05, 0.02, 0.1, 0.5, 1.0, 2.0, and 4.0 μg/mL	1 h	↓ Prog and total mot	Selvaraju et al., 2011 [161]
					↓ Kinematics	
					↑ Tail abnormality	
					↓ Plasmalemma, functional membrane integrities and acrosomal integrities	
					↓ Normal nuclear morphology	
					↑ Nuclear chromatin decondensation	
					↓ MMP	
					↓ Sperm–zona binding and in vitro	
Duroc boar *(Sus scrofa)*	Endogenous	Atrazine, fenoxaprop-ethyl, malathion, and diazinon	50, 100, and 500 μM	1 h	↓ Viability	Betancourt and Reséndiz, 2006 [162]
					↓ Prog mot	
					↓ Kinematics	
Human *(Homo sapiens)*	Endogenous	PCB126, PCB118, and PCB153	2–20 μg PCB/mL	5 hr	No effects observed on sperm mot, vitality, spontaneous AR, or inducibility of the AR.	Pflieger-Bruss et al., 2006 [164]
Human *(Homo sapiens)*	Endogenous	Roundup	1 mg/L	1 and 3 hr	↓ Prog and total mot	Anifandis et al., 2017 [164]
					↓ MMP	

Abbreviations: ↓, decrease or negatively affected; ↑ increased or positively affected; AR, acrosome reaction; ATP, adenosine triphosphate; DNA, deoxyribonucleic acid; h, hour; min, minutes; MMP, mitochondrial membrane potential; mot, motility; PCB, pentachlorobiphenyl; prog, progressive motility; sec, seconds.

## Data Availability

Data are contained within the article. The data presented in this study are available in Table 1 and Table 2.

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
