# Peer review of "Contaminants of Emerging Concern (CECs) and Male Reproductive Health: Challenging the Future with a Double-Edged Sword"

_toxics, 2023, doi:10.3390/toxics11040330_

Round 1
Reviewer 1 Report
The work is very well done and also interesting, in my opinion it can be published after making some revisions:
1) All the materials and methods part are missing, it would be useful to add a paragraph to explain the method used to select the used literature etc.
2) In the part called "impact of CECs on offspring and potential mechanisms" since there is no part clearly dedicated to discussion, the authors could try to better clarify how much the various substances analyzed are actually harmful for humans given that most of the studies analyzed and cited are made on animals models using high quantities of substances that it is difficult for a human being to introduce for example with normal nutrition.
3) As a last suggestion, it might be interesting to investigate (in the authors' favorite section) how some molecules interfere with spermatogenesis and reproduction, such as Bisphenols and PFAS for which there are many studies available in the literature
Author Response
Response to Editor and Reviewer comments on ijerph-2249407
The authors would like to thank the editor for the opportunity to address the concerns of the reviewers and resubmit our manuscript for possible publication. All changes are highlighted as track changes in the revised manuscript and the reviewers’ comments are addressed in detail below. Where applicable, the line numbers are mentioned to indicate where the changes were made in the manuscript.
Reviewer #1
Comments:
“The work is very well done and also interesting, in my opinion it can be published after making some revisions”:
1) All the materials and methods part are missing, it would be useful to add a paragraph to explain the method used to select the used literature etc.
Response: As this is a narrative review, there is no formal structure as seen in a systematic review, and thus the reason why we did not include a definite subsection dedicated to Materials and Methods. “Narrative reviews are publications that describe and discuss the state of science on a specific topic or theme from a theoretical and contextual point of view with little explicit structure for gathering and presenting evidence.” - Henry et al., 2018. We have, however, amended the last paragraph of the manuscript’s Introduction to now comprise of a detailed paragraph describing the reason for the selection of the CECs discussed in the review. In addition, we outline in detail the objectives of the narrative review.
Lines 85-97: “There are three objectives of this narrative review: (i) to present an overview of the existing literature regarding the effects of the selected CECs on male fertility; (ii) to assess the impact of CECs on the offspring of exposed parents; and (iii) to explore the idea of using spermatozoa as a bioindicator of environmental change as well as its potential use for future toxicology studies. Six CECs discussed in this narrative review [pesticides (atrazine, simazine, chlorpyrifos); pharmaceuticals (diclofenac, naproxen and sulfamethoxazole)] were selected based on their high prevalence in the near‐shore marine environment of False Bay around Cape Town [28,29] as well as in rivers [30,31], ambient air [32,33], household dust [34] and human samples in agricultural areas in the Western Cape, South Africa [35]. The effects of the selected CEC groups on male reproductive health were evaluated through a comprehensive survey of in vivo and in vitro studies across taxa, with emphasis on the two groups of contaminants. Overall, this review underscores the im-portance of understanding the potential impact of CECs on male reproductive health.”
2) In the part called "impact of CECs on offspring and potential mechanisms" since there is no part clearly dedicated to discussion, the authors could try to better clarify how much the various substances analyzed are actually harmful for humans given that most of the studies analyzed and cited are made on animals models using high quantities of substances that it is difficult for a human being to introduce for example with normal nutrition.
Response: A clear distinction has now been made between animal and human studies in the revised manuscript. The following sentence has been added in this regard.
Lines 490 - 492: “Whilst evidence in humans is often technically challenging to obtain, in animal models, on the other hand, evidence is mounting and emphasizes that the perturbation of epigenetic marks might account for infertility cases and diseases in the offspring.”
In addition, evidence was provided for the effects of the compounds tested in empirical studies. We have also highlighted the physiological impact of CEC exposure on the male reproductive system, and provided an up to date molecular work which confirmed the molecular signatures left by CEC exposure (see lines 488-546).
3) As a last suggestion, it might be interesting to investigate (in the authors' favourite section) how some molecules interfere with spermatogenesis and reproduction, such as Bisphenols and PFAS for which there are many studies available in the literature.
Response: As indicated in the introduction (lines 85 – 106), we focused mainly on the containments that were detected in the marine environment of False Bay, Cape Town at high concentrations. We have however as requested, mentioned the suggested contaminates in brief.
Lines 501 – 504: “In utero exposure to other forms of CECs including the pesticide permethrin, insect repellent N,N-Diethyl-meta-toluamide, plastic additives such as bisphenol A and phthalates, dioxin, and jet fuel was also shown to promote male germline alterations in F3 progeny [134].”

Reviewer 2 Report
The manuscript deliberated several CECs and their potential effects on male fertility and the offspring of exposed parents, as well as the use of spermatozoa in toxicological studies. The results are promising. However, some revisions are necessary.
1. According to the author's description, CECs in the paper has an impact on semen quality, so whether there is a safe limit for people to control the concentration of CECs.
2. Does CECs affect male reproduction temporarily or permanently? Are conventional treatments effective?
3. How to eliminate or minimize the impact of existing CECs on people?
4. Can the authors meta-analyze existing studies to get stronger evidence?
Author Response
Response to Editor and Reviewer comments on ijerph-2249407
The authors would like to thank the editor for the opportunity to address the concerns of the reviewers resubmit our manuscript for possible publication. All changes are highlighted as track changes in the revised manuscript and the reviewers’ comments are addressed in detail below. Where applicable, the line numbers are mentioned to indicate where the changes were made in the manuscript.
Reviewer #2
Comments:
“The manuscript deliberated several CECs and their potential effects on male fertility and the offspring of exposed parents, as well as the use of spermatozoa in toxicological studies. The results are promising. However, some revisions are necessary.”
1) According to the author's description, CECs in the paper has an impact on semen quality, so whether there is a safe limit for people to control the concentration of CECs.
Response: Although recommended limits have been published for some of the CECs covered in our review (quote references), many of these contaminants have not been investigated to the extent that safe limits have been determined or published.
Lines 307-309:” Despite being banned in the European Union and restricted in other countries, atrazine is still found in water at levels exceeding recommended limits (US Environmental Protection Agency (EPA) = 3 μg/L, European Union = 0.1 μg/L) [57].”
2) Does CECs affect male reproduction temporarily or permanently? Are conventional treatments effective?
Response: An additional section has been added to the manuscript to address the reviewers’ questions (see Section 5). The CECs effects will depend on the time of exposure in an organism’s life in addition to the duration of exposure and nature of the CEC, therefore each CEC on its own has variable effects. With regard to conventional treatments, some investigations have explored the addition of compounds such as antioxidants in combination with a contaminant which is known to induce ROS, thereby exploring the effects of the antioxidants as a possible precautionary treatment to alleviate the effects of CEC exposure. As CEC exposure usually presents as a combination of contaminants instead of one individual contaminate, treatment methods will thus be problematic as various target effects need to be considered. Furthermore, there are conflicting results in the literature as to whether treatment such as antioxidant supplementation could indeed alleviate CEC exposure effects.
Lines 535 – 584: See section 5 - “Management, detection and possible treatment of CEC exposures”.
3) How to eliminate or minimize the impact of existing CECs on people?
Response: Since various CECs include commonly used products in households and agriculture, eradication of the contaminates is unlikely. Thus, the only option to control and minimize the impact of CECs on human health and the environment is to implement regular and accurate detection techniques for the CECs and to further provide strict regulation methods such as legislative laws which control the levels of usage and implement penalties in cases where the CECs are irresponsibly used and discarded.
Lines 578 - 584:” Control measures and mitigation strategies should therefore focus on limiting human exposure to CECs through appropriately identifying and eliminating the compounds from the possible sources as well as imposing penalties for using such compounds in agriculture and manufacturing, especially when alternatives are available. In addition, establishing informational campaigns to increase public awareness about the emergence of CECs and their consequences has the promise to drive legislative action and limit the re-productive effects currently reported.”
4) Can the authors meta-analyze existing studies to get stronger evidence?
Response: We decided to present the manuscript as a narrative review discussing the selected CECs as mentioned in the introduction in addition to the three main objectives. A meta-analysis would require both the structure and points/objectives of the current narrative review to be completely altered, which the authors feel is unnecessary and could be the aim of a future publication.

Round 2
Reviewer 1 Report
Dear authors, after the implementations you have made, the manuscript has certainly improved. Surely it can be said that it is a well-structured work and that it will be able to make an excellent contribution to the field.
Best regards